# Genetics of Diabetic Retinopathy, a Leading Cause of Irreversible Blindness in the Industrialized World

**DOI:** 10.3390/genes12081200

**Published:** 2021-07-31

**Authors:** Ashay D. Bhatwadekar, Aumer Shughoury, Ameya Belamkar, Thomas A. Ciulla

**Affiliations:** 1Department of Ophthalmology, Eugene and Marilyn Glick Eye Institute, Indiana University, Indianapolis, IN 46202, USA; abhatwad@iupui.edu (A.D.B.); ashughoury@gmail.com (A.S.); ameyabelamkar10@gmail.com (A.B.); 2Department of Internal Medicine, Ascension St. Vincent Hospital, Indianapolis, IN 46260, USA; 3Clearside Biomedical, Inc., Alpharetta, GA 30005, USA; 4Retina Service, Midwest Eye Institute, Indianapolis, IN 46290, USA

**Keywords:** diabetic retinopathy, genetics, candidate gene studies, linkage studies, GWAS

## Abstract

Diabetic retinopathy (DR) is a chronic complication of diabetes and a leading cause of blindness in the industrialized world. Traditional risk factors, such as glycemic control and duration of diabetes, are unable to explain why some individuals remain protected while others progress to a more severe form of the disease. Differences are also observed in DR heritability as well as the response to anti-vascular endothelial growth factor (VEGF) treatment. This review discusses various aspects of genetics in DR to shed light on DR pathogenesis and treatment. First, we discuss the global burden of DR followed by a discussion on disease pathogenesis as well as the role genetics plays in the prevalence and progression of DR. Subsequently, we provide a review of studies related to DR’s genetic contribution, such as candidate gene studies, linkage studies, and genome-wide association studies (GWAS) as well as other clinical and meta-analysis studies that have identified putative candidate genes. With the advent of newer cutting-edge technologies, identifying the genetic components in DR has played an important role in understanding DR incidence, progression, and response to treatment, thereby developing newer therapeutic targets and therapies.

## 1. Background

Diabetic retinopathy (DR) and diabetic macular edema (DME) are the leading causes of blindness among diabetic individuals. Based on clinical characteristics, DR is divided into two broad categories: non-proliferative DR (NPDR) and proliferative DR (PDR). NPDR is an early stage of DR associated with an increase in vascular permeability and capillary occlusion. At this stage, the fundus examination exhibits pathologies such as microaneurysms, hemorrhages, and hard lipid exudates. PDR is a more advanced form of DR characterized by new and abnormal vessels that can grow into the vitreous scaffold and then shear with ensuing vitreous hemorrhage and abrupt loss of vision, especially after vitreous traction. Ultimately, this neovascularization can fibrose on the inner retinal surface and contract, leading to tractional retinal detachment. Another complication of DR is DME, which can occur at any stage and is categorized by thickening of the macula due to permeability-mediated sub-and intra-retinal accumulation of fluid. DME causes decreased visual acuity and central visual distortion (metamorphopsia) [1,2]. 

From 2015–2019, the prevalence for DR was 27% among patients with diabetes worldwide, including 25.2% for NPDR and 1.4% PDR, and DME was diagnosed in 4.6% of diabetic patients. The highest prevalence was reported in the Western Pacific region (36.2%), while other areas showed variation in its incidence: the Middle East and North Africa (33.8%), Europe (20.6%), and Asia (12.5%) [3]. It is estimated that DR is present in most subjects with type 1 diabetes (T1D) and nearly 60% of individuals with type 2 diabetes (T2D) after 20 years of living with diabetes. In 2010, an estimated 7.7 million Americans were diagnosed with DR, which is expected to increase to 14.6 million by 2050 [4]. Hispanic Americans are at higher risk of developing DR with a prevalence of 8%, and black and white races show about 5% prevalence. Gender also plays some role in that more males are affected by DR than females in the USA [5].

## 2. Pathogenesis of DR

Numerous changes occur in the retinal vasculature with DR long before fundus examination reveals any pathological changes. Long-standing diabetes leads to an early and progressive loss of retinal pericytes and endothelial cells, resulting in microaneurysms [6]. Furthermore, increased leukostasis and platelet-fibrin thrombi changes in the retinal capillaries contribute substantially to retinal ischemia and vascular obstruction. Resultant chronic hypoxia triggers angiogenic growth factors such as vascular endothelial growth factor (VEGF) that ultimately contribute to the retinal neovascularization associated with PDR [7]. 

Hyperglycemia and chronic inflammation predominantly drive the onset and pathogenesis of DR (Figure 1). Some studies suggest more rapid DR progression with fluctuation in glucose levels between hyper and normoglycemia in individuals with poorly controlled diabetes [8]. Various mechanisms play an integral role behind the capillary damage observed in DR, including (i) increase in polyol pathway, (ii) activation of protein kinase C, (iii) accelerated increase in non-enzymatic glycation and (iv) production of reactive oxygen species (ROS). The activation of the polyol pathway results in a loss of retinal capillary cells. An antioxidant defense system that protects the cell and tissue from oxidative stress is downregulated, specifically the enzymes superoxide dismutase (SOD) and glutathione peroxidase (GSH). An accelerated increase in advanced glycation end products (AGEs) may provoke sustained cell activation, basement membrane damage, and basement membrane thickening, further exacerbating damage to the retinal tissue.

Changes in retinal blood flow patterns also play a vital role in DR pathogenesis, such as (i) increased retinal blood flow and (ii) increased heterogeneity in its distribution [9,10]. Typically, blood flow control is balanced between endothelium-derived relaxing factors, such as nitric oxide (NO), prostacyclin, endothelium-derived hyperpolarizing factor (EDHF), and contracting factors angiotensin II, endothelin, and cyclooxygenase. The interplay of these factors plays a vital role in DR by regulating vascular tone via the stimulation or inhibition of smooth muscle cells and pericytes [11]. Notably, NO is a potent vasodilator, causing smooth muscle relaxation, and hyperglycemia results in either diminished production or quenching of NO by various mechanisms [12]. Prostacyclin (PGI2) works complementary to NO and causes relaxation of smooth muscles through cyclic adenosine monophosphate (cAMP) production, which inhibits calcium-mediated contraction.

Finally, DR is associated with increased VEGF levels in the vitreous as well as the retina. VEGF plays a vital role in neovascularization, resulting in the progression of NPDR to PDR. VEGF not only stimulates endothelial cell migration and proliferation but also acts as a survival factor for new blood vessels in diabetes [13]. 

## 3. The Genetic Basis of DR

### 3.1. Genetics in the Prevalence of DR

Several studies suggest variations in DR prevalence depending on racial background. Harris et al. (1998) found the prevalence of DR in patients with T2D to be 46% higher in non-Hispanic blacks and 84% higher in Mexican Americans than non-Hispanic whites [14]. Notably, the prevalence of diabetes has been found to be highest among American Indians and Alaskan Natives, followed by Hispanics and non-Hispanic whites [15]. As per the latest study involving 53,998 individuals, the prevalence of DR among American Indians and Alaskan Natives was lower than earlier statistics: NPDR (17.7%), PDR (2.3%), DME (2.3%), and sight-threatening retinopathy (STR) (4.2%) [16]. Furthermore, DR’s prevalence in Pima Indians with T2D has been found to be exceptionally high (37.8%) [17]. Similarly, a higher prevalence of DR has been reported in Navajo and Hopi Indians (40%) and Sioux Indian Tribes (45.3%) [16]. In another study involving Oklahoma Indian tribes, the prevalence of DR was found to be 20.1% [18]. These disparities in the prevalence of DR are independent of blood glucose control [19], suggesting the potential genetic basis for DR pathogenesis. 

### 3.2. Familial Risk of DR

Interestingly, the Diabetes Control and Complications Trial (DCCT) highlighted the familial clustering of DR in the families of 372 subjects with 467 first-degree relatives with T1 and T2D. The risk of severe retinopathy was significantly higher among families of individuals with retinopathy. The correlation of severity of retinopathy was higher among parent-offspring; however, a similar familial correlation was not observed for nephropathy [20], suggesting the influence of an independent genetic risk factor. 

The Find-Eye study provides further insights into the inherited factors that cause DR’s susceptibility and severity. This study involved 2368 diabetic subjects from 767 families and showed that the heritability of DR was 27%, with a polygenic heritability of PDR of 25% [21]. Hietala et al. additionally provided evidence that clustering exists for PDR in families with two or more siblings with T1D, demonstrating an increased risk of DR in siblings of probands with DR, after adjusting for glycated hemoglobin, duration of diabetes, and mean blood pressure [22]. Finally, twin studies have also reported the same degree of DR severity among twins with DM, with one early study demonstrating a concordance of 65% among T1D and 95% among T2D twins [23]. Thus, the severity of DR is apparently influenced by familial factors.

### 3.3. Progression of DR

DR is known to follow the classical course of disease progression from no-DR→NPDR→PDR; however, not every patient progresses through this classical paradigm. For example, some patients develop mild NPDR after 8-10 years of no DR. Of these individuals, only 50% of those with T1D develop PDR, and 20% with type 2 diabetes develop PDR. After four years of moderate to severe NPDR, the incidence of PDR is 11%, and the incidence of severe vision loss is 7.2% [9]. It is unclear why only a certain percentage of people develop PDR while the remaining are protected from DR despite a longer duration of diabetes. Previously, we identified a similar population of individuals who remained protected from DR in spite of long-standing diabetes. We performed microarray studies on circulating angiogenic cells (CACs) obtained from this population to demonstrate mapping with unique targets such as ataxia telangiectasia mutated (*ATM*) [24], transforming growth factor (*TGFB1*), plasminogen activator inhibitor (PAI-1; gene *SERPINE1*), endothelial nitric oxide synthase (eNOS; gene *NOS3*), and angiotensin-converting enzyme 2 (ACE2) [25]. Our study likely suggests that genetic mechanisms play a role in DR progression.

## 4. Genetic Studies

### 4.1. Candidate Gene Studies 

Candidate gene studies are clinical and preclinical studies where a gene is identified as potentially implicated in disease pathogenesis based on the expression of already identified proteins in the disease state. A 2009 meta-analysis examined 34 genetic variants known to be associated with the pathogenesis of DR and found the aldose reductase (ALR) gene aldo-keto reductase family 1 member B, *(AKR1B1*) to have the highest number of polymorphisms associated with DR irrespective of ethnicity. Additional polymorphisms reported to be significantly associated with DR included *NOS3*, *VEGFA*, integrin subunit alpha 2 (*ITGA2*), and intercellular adhesion molecule 1(*ICAM1*) [26]. More recently, Sharma et al. (2019) elegantly summarized candidate gene studies associated with DR in a review highlighting 65 different genes [27]. These candidate genes are involved in a variety of cellular processes, such as glucose metabolism, inflammation, angiogenesis, and neurogenesis. These studies have overall yielded inconsistent and controversial results [28].

The majority of these studies have been performed in T2D; however, some targets were associated with T1D in candidate gene studies, such as peroxisome proliferator-activated receptor gamma (*PPARG*), neuropeptide Y receptor 2 (*NPY2R*), centrosomal protein (CEP)-135 (*CEP135)*, and plexin domain containing 2 (*PLXDC2*). The following sections focus on those genes that have been found to be associated with DR based on multiple independent studies. Additionally, Table 1 summarizes the findings of these studies. 

#### 4.1.1. Aldose Reductase (ALR) 

Aldose reductase has been significantly implicated in the pathogenesis of DR via the polyol pathway, by which excess intracellular glucose in retinal cells is converted to fructose [46]. The ALR enzyme catalyzes the initial, rate-limiting step of the pathway, which consists of NADPH-mediated glucose reduction into sorbitol. Sorbitol cannot diffuse freely from the cell cytoplasm; while initially it was suggested that sorbitol accumulation leads to osmotic stress [47], the amount of sorbitol measured in diabetic vessels and nerves was found to be insufficient to cause osmotic stress [48]. Furthermore, increased ALR and polyol pathway activity has also been shown to be significantly associated with pericyte degeneration, a key feature of DR, leading to abnormalities of vascular permeability and compromise of the blood-retinal barrier. In turn, breakdown of the blood-retinal barrier is thought to be a major etiology for the development of exudates and macular edema in DR [49].

The *AKR1B1* gene is located on chromosome 7q33 and is expressed in retinal capillary pericytes to produce ALR. The three most reported *AKR1B1* polymorphisms associated with DR include (i) rs759853 (106C/T) [50], (ii) 5′-(CA)n microsatellite (5′*AKR1B1*) [51], and (iii) rs9640883 [26,52]; however, studies of the association between these SNPs and DR have frequently resulted in inconsistent and contradictory findings. 

Several studies of the rs759853 single nucleotide polymorphisms (SNPs) have demonstrated that the C allele is significantly associated with increased risk of DR, while the T allele may be protective [26,29,50,53,54,55,56,57,58,59]. This finding was replicated by a recent 2020 meta-analysis of 23 studies [60]. Interestingly, two recent studies of T2D in Eastern European and Arab populations demonstrated the opposite relationship, suggesting that the T allele is associated with increased risk and severity of DM, while the C allele is protective [61,62]. Finally, some studies have found no such association [63,64,65,66], including a 2018 meta-analysis [29]. It is possible that some of the variation in these findings is due to the racial heterogeneity of the studies in question; however, when stratifying by type of diabetes, the T allele of the SNP has been found to be protective against the development of DR in T1D regardless of the ethnic background [26,29]; therefore, differences in the pathogenesis of DR in T1D compared to T2D may also account for the heterogeneity of these findings.

Microsatellites are tandem repeats of 1–6 bp that are widely spread throughout the genome. The (CA)n microsatellite has also been studied in relation to DR pathogenesis, with similarly diverse findings. The three most commonly studied alleles of this SNP in DR are z, z+2, and z-2. Some studies have suggested that the z-2 allele is associated with increased ALR activity [67] and DR pathogenesis in both T2D (stronger association) and T1D (weaker association) patients [50,53,68,69]. Two large meta-analyses have confirmed these findings, and both additionally found that the z+2 allele may be significantly protective against DR development in T2D [26,70]. Notably, however, Abhary et al. (2010) found no significant association between any (CA)n allele and T2D once established risk factors were controlled for [30], suggesting that the (CA)n microsatellite may be only secondarily associated with DR through other clinical risk factors, rather than playing a primary role in its pathogenesis.

Further, the rs9640883 has also been associated with DR pathogenesis. Specifically, the C allele has been implicated in an increased risk of DR, while the T allele has been associated with decreased risk [30,57,58,61]; however, while Abhary et al. (2010) reported a significant association of rs9640883 with DR in general, they did not find this association to be significant after adjusting relevant clinical factors except for the duration of diabetes. The authors, therefore, suggest that the association of the rs9640883 allele with DR may be attributed largely to younger age at onset of diabetes [30], suggesting that this allele may be associated with the early development of diabetes as a confounding variable, rather than affecting the pathogenesis DR itself.

Other *AKR1B1* SNPs that have been studied in DR include rs5053, rs918825, and rs706207; however, these were not found to be significantly associated with DR in a recent meta-analysis [71].

#### 4.1.2. Receptor for Advanced Glycation End Products (RAGE)

A major consequence of hyperglycemia is excessive non-enzymatic glycation of proteins and other macromolecules. French chemist L.C. Maillard first reported this reaction in 1912. In the Maillard reaction reducing sugars such as glucose react with an amino group of proteins to form irreversibly modified proteins called advanced glycation end products (AGEs) [72]. AGEs accumulate both intracellularly and extracellularly and may significantly modify proteins, thereby altering function [73]. Diabetes enhances AGE accumulation in the retina’s vascular and neural components throughout life [74]. AGEs could affect the physiological function of pericytes [75] and may even lead to cell death [76]. The receptor for AGE (RAGE) was first observed on the endothelial cell surface [77] and among one of the well characterized receptors of AGEs. AGE interactions with RAGE trigger pro-inflammatory cascades, such as P21 (Ras)/MAP kinase, and AGE is associated with decreased production of prostacyclin (PGI2) and an increase in platelet aggregation and fibrin stabilization [78]. AGE binding to RAGE receptors upregulates cyclooxygenase-2 (COX), resulting in monocyte activation and vascular cell dysfunction [79], linked to oxidative stress and NF-κB activation [80,81]. Finally, AGEs have been shown to stimulate endothelial cell proliferation and angiogenesis by upregulation of VEGF and via the RhoA/Rho-associated protein kinase pathway [82,83].

Several polymorphisms in the gene encoding RAGE (*AGER*) have been identified as potentially associated with DR: rs1800625 (429T/C), rs1800624 (374T/A), rs2070600 (Gly82Ser), rs184003 (1704G/T), and rs3134940 (2184A/G). The rs1800624 and rs1800625 SNPs are perhaps the most widely studied. These were initially identified by Hudson et al. (2001), who found the C allele of the rs1800624 SNP to be significantly associated with DR in D2M subjects, while the A allele of the rs1800624 SNP was elevated, but not statistically significantly associated with DR. The association between the rs1800624 SNP has since been replicated in Indonesian subjects [84]; however, this relationship was not found in a 2012 meta-analysis by Yuan et al. [85].

Lindholm et al. (2006) further studied the rs1800624 SNP in Scandinavian T1D and T2D patients. The rs1800624 polymorphism was found to be related to different major histocompatibility complex, class II, DQ beta 1 (HLA-DQB1) genotypes and associated with the presence of DR in Scandinavian individuals, with a higher frequency of the A allele in T1D than T2D individuals with DR [31]. Similar findings have been more recently replicated in Pakistani individuals [86]. Furthermore, two recent meta-analyses (2016 and 2017) have provided additional evidence that the A allele of the rs1800624 polymorphism is associated with significantly increased risk of DR in both Asian and Caucasian diabetic populations [87,88], though in the case of Yu et al. 2016, this association was not sustained after T2D patients were excluded from the analysis. Conversely, two earlier meta-analyses (2010 and 2012) suggested a protective effect of the A allele in Caucasian patients with D2M [85,89]. Finally, no significant association was observed between either the rs1800624 or rs1800625 polymorphisms in a study of Malaysian patients [90] or Indian patients [32].

The rs2070600 polymorphism was also assessed in an Indian population with DR. The *AGER* polymorphism for rs2070600 was found to be significantly associated with DR [32,91]. This finding has been since replicated in Indonesian patients [84] and Han Chinese patients [92]. A 2002 meta-analysis has also validated the association of the Gly82Ser polymorphism with DR in a general Asian population [88]; however, this finding was only supported by three later meta-analyses (2012, 2012, and 2016) after assuming a genetic recessive model [85,88,93]. Finally, the rs184003 RAGE polymorphism has also been reported to be associated with DR in Japanese and East Asian populations [94,95].

While some of the above studies provide encouraging evidence for the association of the AGER gene with DR, other studies have not supported an association between RAGE polymorphisms and DR. As noted above, it is worth emphasizing that two 2012 meta-analyses found no significant association between rs1800624 or rs184003 and DR, despite stratification by ethnicity [85,93]. A subsequent 2016 meta-analysis moreover similarly showed no significant association between rs1800624 and DR [88]. Finally, while one study of rs184003 showed a significant association with DR in Japanese subjects as above, other studies have failed to identify a similar association [96,97]. Other SNPs such as rs3134940 have similarly not been found to be associated with DR pathogenesis [97]. 

#### 4.1.3. *VEGF* Gene

VEGF is a potent cytokine known to play an integral role in DR pathogenesis by altering retinal vascular permeability, promoting endothelial dysfunction, and ultimately inducing retinal angiogenesis [13]. VEGF is a part of the growth factor family that includes VEGF-A, VEGF-B, VEGF-C, VEGF-D, VEGF-E, VEGF, and placental growth factor (PGF). VEGF receptor 2 (VEGFR2), also known as KDR/Flk-1, mediates vascular leakage and angiogenesis. While VEGF-A binds to VEGFR2 and VEGFR1 (Flt-1), VEGF-B and PGF binding are more specific to VEGFR1. VEGF-C and VEGF-D bind to VEGFR2 and VEGFR3 (Flt4) [98]. Recent meta-analyses have demonstrated that serum VEGF levels are significantly associated with both the presence and severity of DR in diabetic patients [86,99]. Additionally, intravitreal (IVT) injection of anti-VEGF-A is the first-line treatment for DME. Currently available anti-VEGFA medications for DME include ranibizumab, off-label bevacizumab, and aflibercept, the latter of which also binds VEGF-B and PGF [1,100]. 

The human *VEGFA* gene is present on chromosome 6 (6p21.1) encodes eight exons, while other human *VEGF* genes encode seven exons. Alternative splicing of the *VEGF* gene results in at least six transcripts containing exons 1-5 and 8. Additionally, diversity is generated via alternative splicing of exons 6 and 7. *VEGF* gene is highly polymorphic [101], and these polymorphisms are mainly located in the 5′ untranslated (UTR) region and in the 3′ UTR region [102]. *VEGF* polymorphism of rs699947, rs3025039, and rs833061 has been associated with the risk of NPDR among Asians, and rs2010963) has been associated with the risk of NPDR among Caucasians [33,103]. Moreover, rs69947, rs2010963, and rs3025039 have specifically been shown to increase *VEGF* mRNA and protein levels, driving the pathogenesis of DR [104]. Similarly, the DCCT/Epidemiology of Diabetes Interventions and Complications (EDIC) study discovered SNP for rs3025021 in T1D. Further, the family-based analyses demonstrated excess transmission of C at rs699947, T at rs3025021 and the C-T haplotype for both SNPs [105]. Some of the earlier studies by Abhary et al. assessed multiple tag SNPs for VEGF-A polymorphism, demonstrating a significant association with DR, including the AA genotype of rs699946 and the GG genotype of rs833068 for T1D as well as the C allele of rs3025021 and the G allele of rs10434 for T2D [106].

While there is mounting evidence of *VEGFA* polymorphism associations in DR, there are also regional and ethnic variations regarding VEGF, *VEGFA* polymorphisms, and DR. A 2019 meta-analysis of 29 previous studies found serum VEGF levels to be associated with DR in East Asian populations, but not European populations. A recent study by Khan et al. assessed SNPs rs833061, rs13207351, rs1570360, rs2010963, rs5742632, and rs6214 in NPDR, PDR, and control populations of Pakistani origin. This study could only find an association of rs13207351 with NPDR and rs5742632 with PDR [107]. Similarly, studies performed in the Caucasian Slovenian and Mexican PDR subjects did not observe an association between AA genotype of rs6921438 or rs10738760 [108,109] and rs3025021, rs3025035, rs2010963, respectively [110]; however, recent studies have also shown that rs2010963 is associated with the development and severity of DR in Egyptian [111] and Sudanese [112] patients.

Studies of the rs699947 SNP have also yielded varying results across ethnicities. Some studies have demonstrated that the rs699947 SNP is associated with DR in various ethnicities, including Korean [113], Chinese [114], Japanese [115], Pakistani [86], Australian [106], and Spanis [116] populations. However, other studies in Egyptian [117] and Chinese [118] populations have failed to find such an association, including two meta-analyses (2014 and 2015), which analyzed patients pooled from both European and East Asian population studies [104,119]. Thus, while it appears that significant regional and ethnic variations in VEGF polymorphisms and expression may play a confounding role in ethnically heterogeneous studies, the precise nature of these ethnic variations remains unclear.

#### 4.1.4. eNOS Gene

Nitric oxide (NO) is a vasodilatory gas that plays a key role in the regulation of retinal vascular diameter [120]. Endothelial nitric oxide synthase (eNOS; gene *NOS3*) is the major enzyme responsible for the synthesis of nitric oxide (NO) in human vasculature [121]. eNOS also functions to maintain a protective anti-proliferative and anti-apoptotic environment within the vasculature [122,123]. Knockout eNOS^−/−^ mice have been found to develop retinal endothelial dysfunction [124] and demonstrate accelerated development of retinopathy [125], suggesting that eNOS dysfunction may play a role in the pathogenesis of DR. Moreover, NO has been found to be significantly elevated in the vitreous of patients with proliferative diabetic retinopathy [126].

The *eNOS* gene is located in chromosome 7q36.1. The polymorphisms of eNOS rs869109213 (4b/a) and rs2070744 (-786T/C) have been found to be associated with DR in Tunisian [34] and Algerian [127] populations. A multi-ethnic meta-analysis by Zhao et al. (2012) also demonstrated a protective effect associated with the C allele of the rs2070744 SNP. Additionally, the 4a/4a genotype of the 27VNTR SNP has also been associated with a lower risk of developing DR [128], suggesting a protective effect of the 4a allele. This finding was replicated by Zhao et al. (2012) across a range of European and Asian ethnicities [129] and again in African patients, specifically by Qian-Qian et al. (2014) [130]. On the other hand, Cilenšek et al. (2012) found that the 4a/4a genotype is instead associated with an increased risk of PDR in Caucasian patients [131]. Lastly, multiple cross-sectional studies and meta-analyses have not found any association between the 27VNTR SNP and DR [132,133]. Most recently, Momeni et al. (2016) have suggested that the 27VNTR SNP may be associated with the development of T2D but not DR itself [134], which may explain some of these disparate findings.

#### 4.1.5. Angiotensin-Converting Enzyme Gene

Angiotensin-I converting enzyme (ACE) is a part of the renin-angiotensin system (RAS) regulating vascular tone. ACE converts angiotensin I to angiotensin II, which in turn causes vasoconstriction. Increased ACE activity, therefore, decreases retinal blood flow and can lead to pathologic angiogenesis [135]. The RAS may be implicated in the mechanism by which VEGF induces vascular permeability [136]; therefore, several studies have suggested that ACE inhibitors and angiotensin receptor blockers may reduce the risk of DR in diabetic patients [137,138,139]. Recently, tears of patients with DR have been found to have elevated ACE and angiotensin II concentration [140], further implicating ACE expression with DR.

The *ACE* gene is located on the 17q23.3 chromosome. Most of the phenotypic variance in this gene’s expression is due to the intron 16 insertion/deletion (I/D) polymorphism (rs1799752 or also rs4646994) [141]. The I/D polymorphism is the most widely studied in association with DR. In their 2016 meta-analysis, Luo et al. found this polymorphism to be significantly associated with DR [35]; however, other studies have failed to find the association of I/D polymorphism with DR [142,143,144,145]. Additionally, the ACE rs4343 (2350G/A) polymorphism has also been reported to be associated with DR [36].

#### 4.1.6. Erythropoietin Gene

While initially, erythropoietin’s (EPO) role was thought to be limited to the regulation of erythrocyte production in bone marrow, emerging studies demonstrate the pivotal role of EPO in various bodily functions, including the retina. In the early stages of DR, EPO exhibits a neuroprotective, anti-apoptotic, anti-inflammatory, anti-oxidative, and anti-permeability action by protecting retinal pigment epithelium against increased permeability. Although several preclinical and small interventional clinical studies show the potential benefit of intraperitoneal or intravitreal EPO administration in early DR [146], there is some concern regarding the use of EPO treatment in advanced stages of DR because EPO acts synergistically with VEGF, thus promoting angiogenesis. Elevated serum EPO has therefore been associated with the progression of NPDR to PDR [147] and EPO has been reported in abnormally high concentrations in the vitreous of PDR eyes [126].

Erythropoietin gene *EPO* is located on 7q22.1. There are mixed reports regarding EPO’s polymorphism and DR. While some studies have demonstrated an association between rs551238, rs1617640, and rs507392 and an increased risk of developing PDR [37,38,39], a recent 2020 meta-analysis could not find such an association [148], and an association was only observed between rs551238 and DR in a 2017 meta-analysis study [149]. 

### 4.2. Linkage Studies

Genetic linkage analysis is based on the hypothesis that genes residing in close proximity physically on a chromosome endure linkage during meiosis. Two genetic markers in physical proximity to each other are unlikely to be separated during the chromosomal crossover. Thus, upon identification of a chromosomal location of a particular disease phenotype, the genetic analysis further helps to decipher whether the disease is caused by a mutation in a single gene or there is an involvement of other mutations [150]. Traditionally linkage analysis studies were largely supplemented with robust genome-wide association studies (GWASs); however, with the advent of whole-genome sequencing, linkage analysis is again emerging as a critical tool for disease pathogenesis [151].

Vuori et al. performed genome-wide sib-pair linkage analysis for *CACNB2*, a gene for voltage-dependent L-type calcium channel subunit beta-2, to report linkage 10p12 overlapping the *CACNB2* gene in PDR for T1D subjects. Furthermore, sequencing of *CACNB2* revealed two coding variants, rs202152674 (R476C) and rs137886839 (S502L) [40]. In another study by Hallman et al., a genome-wide linkage scan was performed using 794 diabetes subjects from 393 Mexican American families, involving at least two diabetic siblings. The sample consisted of 567 individuals with retinopathy, including 282 affected sibling pairs. The study used 360 polymorphic markers with an average spacing of 9.4 cM to perform nonparametric linkage analysis. This study reported the highest risk of retinopathy for all families were on chromosome 3 and chromosome 12, while the risk of moderate to severe NPDR occurred on chromosomes 5, 6, and 19 [152].

### 4.3. Genome-Wide Association Studies 

GWASs involve scanning genome for SNPs that are common in the human genome and determines how these polymorphisms are distributed across the different patient populations. GWAS serves a dual-purpose: (i) to uncover associations between individual SNPs and risk of disease between generations and (ii) to identify an individual’s risk for a particular condition [153,154]. In contrast to single-gene studies, GWAS assesses the entire genome; hence, GWAS is a *non-candidate-driven* approach. GWAS cannot specifically identify which genes are causal for a disease. The first successful GWAS study was published in 2002 on myocardial infarction, followed by a landmark study on age-related macular degeneration (AMD) that identified the complement factor H gene as a major risk factor for AMD [155]. As of 2020, the online repository GWAS catalog contains 4795 publications and 222481 associations, including 12 publications and five traits for DR [156].

Grassi et al. performed GWAS studies in individuals with severe DR as defined by DME or PDR in two large cohorts of T1D: (i) genetics of Kidney in Diabetes (GoKinD) and (ii) EDIC. The study included 973 cases and 1856 controls and assessed 2,543,887 SNPs to identify an intergenic SNP association, rs476141, on chromosome 1. This intergenic locus, rs476141, is situated between AKT serine threonine kinase 3 (*AKT3)* and zinc finger protein 238 (ZNF238). These genes play an essential role in cell survival, insulin signaling, and angiogenesis [41]. Interestingly, a similar finding for the role of insulin signaling was reported in a GWAS study performed in a Mexican population where *CAMK4* (calcium/calmodulin-dependent protein kinase IV) was found to be highly associated with DR along with *FMN1* (formin-1) [42]. *CAMK4* increases transcriptional activity required for activating transcription factor-2 (*ATF2*) induced insulin gene expression [157]. 

Burdon et al. used an attractive study design where candidates identified in a GWAS study were subsequently validated in human retinas and animal studies. This study reported genetic variation near growth factor receptor bound-2 (*GRB2)* on chromosome 17q25.1 associated with sight-threatening DR. In human retinas, the GRB2 was observed to be expressed by all the retina layers, including blood vessels [43]. Furthermore, an animal model of DR showed that GRB2 is upregulated. *GRB2* binds phosphorylated insulin receptor substrate 1 and activates the MAPK pathway via Ras in response to insulin. GRB2 is also involved in VEGF signaling [43], further implicating it in the pathogenesis of DR. In a larger multicentric GWAS study involving eight European and seven African cohorts, the C allele of rs142293996 in an intron of nuclear valosin-containing protein-like (*NVL)* was found to be associated with DR in European subjects. The *NVL* gene is expressed widely in the retina and encodes ATPASes associated with the cellular activities (AAA) superfamily [44]. Recently, Imamura et al. performed a GWAS study in Japanese subjects to identify novel SNP loci rs12630354 near STT3 Oligosaccharyltransferase Complex Catalytic Subunit B *(STT3B)* and rs140508424 in paralemmin-2 (*PALM2)* confers susceptibility to DR. This study also identified a novel gene EH domain containing 3 (EHD3) associated with DR susceptibility [45].

## 5. Conclusions and Future Perspectives 

Genetics play an essential role in DR potentially explaining features, such as incidence, progression, heritability, and pathogenesis [158]. Accumulating evidence suggests that DR exhibits some level of inheritance; however, there is no monogenic association. Genetic studies show a racial variation in DR’s incidence. Heritability of DR is estimated as high as 27% and even higher (52%) for a more severe form [158] after adjusting known confounders such as blood pressure glycemic control and duration of diabetes. Putative candidate gene studies have highlighted various candidate genes associated with DR, such as *AKR1B1*, *VEGFA*, *AGER*, *EPO*, and *NOS3*; however, some of these gene candidates failed to replicate in populations with different ethnicities, again highlighting the role of genetics. We acknowledge that our selection of candidate genes is primarily based on the pathogenesis of DR. This review is no exhaustive summary of candidate genes for DR. For instance, a recent DRGen study identified additional candidate genes with PDR such as kruppel like factor 17 (*KLF17**)*, zinc finger protein 395 (*ZNF395*), myeloid cell surface antigen (CD33), pleckstrin homology domain-containing family G member 5 *(*PLEKHG5*)*, and collagen type XVIII alpha 1 chain (*COL18A1*) [28]. Several of these candidate genes are known to have VEGF regulatory functions [28]. Future studies will likely identify the role of these genes in DR using different patient populations. Further, GWAS, a non-candidate-driven approach, may help curtail some limitations of candidate gene studies by uncovering associations between individual SNPs and DR risk. A novel approach of devising a genome wide polygenic score may help to determine the susceptibility of an individual to DR [159]. Together, precision medicine may enhance prevention and treatment strategies using transcriptomic studies. With advancing technologies and newer platforms for genetic analysis in DR, the future for treatment advances is bright.

## Figures and Tables

**Figure 1 genes-12-01200-f001:**
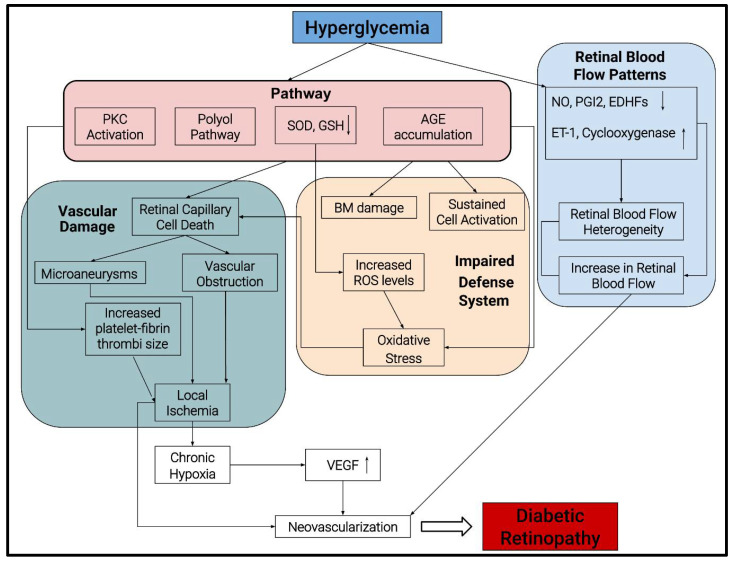
Schematic showing pathogenic mechanisms of diabetic retinopathy (DR).

**Table 1 genes-12-01200-t001:** Candidate gene studies in DR.

Candidate Gene	Gene Location	Online Mendelian Inheritance in Man (OMIM) Entry	Polymorphisms	Effect on DR	References
Aldose Reductase (*AKR1B1*)	7q33	103880	rs759853	Protection from DR in T1D patients	[29]
			Z-2 Z Z+2	Risk for DR in T1 and T2D Protection from DR in T2D	[26]
			rs9640883	Duration of diabetes T1D and T2D	[30]
Receptor for Advanced Glycation End Product (*AGER*)	6p21.32	600214	rs1800624	Risk of DR T1D	[31]
			rs2070600	Risk of DR in T2D	[32]
Vascular Endothelial Growth Factor (*VEGFA*)	6p21.1	192240	rs3025039 rs3025021, rs13207351 rs2146323 rs2010963 rs25648, rs833061 rs2010963	Risk of DR in T2D Risk of DR in T2D Risk of DR in T2D Risk of DR in T2D Risk of PDR in T2D	[33]
Endothelial Nitric Oxide Synthase (*NOS3)*	7q36.1	163729	rs869109213 rs2070744	Risk of DR in T2D	[34]
Angiotensin-I Converting Enzyme (*ACE*)	17q23.3	106180	rs1799752 rs4343	Risk of DR in T2D Risk of DR in T2D	[35,36]
Erythropoietin (*EPO*)	7q22.1	133170	rs551238 rs1617640 rs507392	Risk of DR	[37,38,39]
Calcium channel voltage dependent beta-2 sub unit (*CACNB2*)	10p12.33-p12.31	600003	rs202152674 rs137886839	Increased risk of PDR	[40]
Intergenic locus in between *AKT3* and *ZNF238*	1:24401312		rs476141	Increased risk of DR	[41]
Caclium/Calmodulin-Dependent Protein Kinase IV (*CAMK4*)	5q22.1	114080	rs2300782	Increased risk of DR	[42]
Formin 1 (*FMN1*)	15q13.3	136535	rs10519765	Increased risk of DR	[42]
Growth factor receptor bound-2 (*GRB2)*	17q25.1	108355	rs9896052	Sight threatening DR	[43]
Valosin -containing protein like (*NVL*)	1q42.11	602426	rs142293996	Increased risk of DR	[44]
STT3 Oligosaccharyltransferase Complex Catalytic Subunit B *(STT3B)*	3p23	608605	rs12630354	Increased risk of DR	[45]
Paralemmin-2 (PALM2AKAP2)	9q31.3	604582	rs140508424	Increased risk of DR	[45]

## Data Availability

Not applicable.

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
