# Peer review of "Genetics of Diabetic Retinopathy, a Leading Cause of Irreversible Blindness in the Industrialized World"

_genes, 2021, doi:10.3390/genes12081200_

Round 1

Reviewer 1 Report

This reads like it is from a draft of a chapter written for some sort of thesis. A lot of work clearly went into it, and the first author should be commended for this. However, it was hard to follow one thread through the article. I came across numerous typos and grammatical errors - too many to list. An example is on line 143, but there are several sections, some of which run to a few lines long, that just do not quite make sense. Please consider reading through your writing carefully prior to submission for publication, in order to avoid this problem. There were also many unexplained abbreviations, many overstatements, and there was a general lack of context for much of the work presented. As the data presented was not truly comprehensive, it felt as if it had been cherry-picked, with no clear explanation as to why the authors chose to focus on certain cytokines/polymorphisms/studies. When writing about genetics of diabetic retinopathy I think it would be helpful to distinguish between the different types of diabetes as that would be an obvious confounder. Also, your section on "Incidence of DR" is mostly about prevalence of DR and other unrelated information.

Author Response

We thank the reviewer for suggestions. We thoroughly read the manuscript, the abbreviations are corrected at the first appearance. Our goal for selecting candidate genes was mainly based on diabetic retinopathy (DR) pathogenesis and candidate genes associated with DR in multiple studies. We added the rationale for choosing candidate gene studies in section 4.1 and acknowledged our review's limitation in section 5. We also added candidate gene polymorphisms observed in type 1 diabetes in section 4.1. Additionally, information pertinent to type 1 diabetes can also be found in sections 3.2 (lines119-124), 3.3 (lines 127-131), 4.1 (lines 150-153), 4.1.3 (lines 239-245). We corrected the incidence of DR to the prevalence of DR. We removed additional information and added subsection 3.2 for more clarification.

Reviewer 2 Report

Bhatwadekar et al. present a review of the literature discussing the genetics of diabetic retinopathy. They report on studies that have been key in developing an understanding of the pathogenesis of DR, including its genetic basis. The authors present a summary of the literature relating to candidate gene studies, linking studies and GWAS relating to DR in different ethnic groups.

The manuscript is generally well written and briefly summarises the literature. The following recommendations should be considered:

  1. The candidate genes discussed in this review represent only a small number of the genes and studies published in the literature. It is understandable that the authors have chosen to focus on a particular subset of genes/studies however, the reasoning behind the specific genes or studies chosen needs to be clearly outlined (whether this was related to sample size, ethnicity etc.).

  1. The following grammatical errors should be corrected prior to publication:

Abstract, line 27: should read “to treatment, and the development of newer therapeutic targets and therapies.”

Section 3.1, Line 135-136: should read “The familial risk of PDR was estimated in 168 of 188 sibships after adjusting for glycated hemoglobin, duration of diabetes, and mean blood pressure”

Section 3.2, line 133- 144: this sentence should be reworded “The prognosis of PDR is 11 %, and severe vision loss is 7.2 % after four years of moderate to severe non-proliferative DR [9]. “

Section 4.1, lines 162-163: this sentence is incomplete and should be reworded “Table 1 has summarized some of these candidates, the following is a summary of some top candidate gene studies “

Section 4.1.6, line 304-305: Should read “EPO plays a neuroprotective role in the early stages of DR and protects the retinal pigment epithelium against increased permeability.”

Section 4.3, lines 347-349: should read “GWAS serve a dual-purpose (i) to uncover associations between individual SNPs and risk of disease between generations (ii) to identify an individual’s risk for a particular condition [80, 81].”

There are multiple minor grammatical errors in the conclusion section that should be addressed.

Conclusion, Lines 403-404: this sentence should be reworded “Precision medicine is an emerging area in a standard of care that focuses on developing better prevention and treatment strategies using cutting edge transcriptomics platforms. “

  1. The reference list should be reviewed and notation corrected, particularly relating to studies published by a study group. For example:

Control, D.; Group, C.T.R. Clustering of long-term complications in families with diabetes in the diabetes control and complications trial. Diabetes 1997, 46, 1829.

Author Response

We thank the reviewer for the helpful suggestions. We chose candidate genes based on their involvement in DR pathogenesis and the presence of multiple studies. We added this rationale in section 4.1 and limitations of our review in section 5

  • The following grammatical errors should be corrected prior to publication:

Abstract, line 27: should read “to treatment, and the development of newer therapeutic targets and therapies.”

Corrected

Section 3.1, Line 135-136: should read, “The familial risk of PDRwas estimated in 168 of 188 sibships after adjusting for glycated hemoglobin, duration of diabetes, and mean blood pressure”

Corrected

Section 3.2, line 133- 144: this sentence should be reworded “The prognosis of PDR is 11 %, and severe vision loss is 7.2 % after four years of moderate to severe non-proliferative DR [9]. “

Corrected

Section 4.1, lines 162-163: this sentence is incomplete and should be reworded “Table 1 has summarized some of these candidates,the following is a summary of some top candidate gene studies “

Corrected

Section 4.1.6, line 304-305: Should read “EPO plays a neuroprotective role in the early stages of DR and protects the retinal pigment epithelium against increased permeability.”

Corrected

Section 4.3, lines 347-349: should read, “GWAS serve a dual-purpose (i) to uncover associations between individual SNPs and risk of disease between generations (ii) to identify an individual’s risk for a particular condition [80, 81].”

Corrected

There are multiple minor grammatical errors in the conclusion section that should be addressed.

Conclusion, Lines 403-404: this sentence should be reworded“Precision medicine is an emerging area in a standard of care that focuses on developing better prevention and treatment strategies using cutting edge transcriptomics platforms.

We performed thorough proofreading of the manuscript to correct grammatical errors. The sentence starting with ‘Precision medicine’ is corrected.

The reference list should be reviewed and notation corrected, particularly relating to studies published by a study group. For example: Control, D.; Group, C.T.R. Clustering of long-term complications in families with diabetes in the diabetes control and complications trial. Diabetes 1997, 46, 1829.

The references are formatted as per the MDPI journal format. Unfortunately, we can not correct the journal format.